# Improving Nursing Assessment in Adult Hospitalization Units: A Secondary Analysis

Irene Llagostera-Reverter [1], David Luna-Aleixos [1,2,*], María Jesús Valero-Chillerón [1], Rafael Martínez-Gonzálbez [2], Gema Mecho-Montoliu [2] and Víctor M. González-Chordá [1,3,*]

1 Nursing Research Group (GIENF Code 241), Nursing Department, Universitat Jaume I, 12071 Castellón, Spain; llagoste@uji.es (I.L.-R.); chillero@uji.es (M.J.V.-C.)
2 Hospital Universitario de La Plana, Vila-Real, 12520 Castellón, Spain; martinez_rafgon@gva.es (R.M.-G.); mecho_gem@gva.es (G.M.-M.)
3 Nursing and Healthcare Research Unit (INVESTÉN-ISCIII), Institute of Health Carlos III, 28029 Madrid, Spain
* Correspondence: dluna@uji.es (D.L.-A.); vchorda@uji.es (V.M.G.-C.); Tel.: +34-964-387744 (V.M.G.-C.)

**Abstract:** The main objective of this study was to analyze the impact of a multifaceted strategy to improve the assessment of functional capacity, risk of pressure injuries, and risk of falls at the time of admission of patients in adult hospitalization units. This was a secondary analysis of the VALENF project databases during two periods (October–December 2020, before the strategy, and October–December 2021, after the strategy). The quantity and quality of nursing assessments performed on patients admitted to adult hospitalization units were evaluated using the Barthel index, Braden index, and Downton scale. The number of assessments completed before the implementation of the new strategy was $n = 686$ (28.01%), versus $n = 1445$ (58.73%) in 2021 ($p < 0.001$). The strategy improved the completion of the evaluations of the three instruments from 63.4% ($n = 435$) to 71.8% ($n = 1038$) ($p < 0.001$). There were significant differences depending on the hospitalization unit and the assessment instrument ($p < 0.05$). The strategy employed was, therefore, successful. The nursing assessments show a substantial improvement in both quantity and quality, representing a noticeable improvement in nursing practice. This study was not registered.

**Keywords:** nurses; nursing; nursing assessment; nursing records; hospitalization





## 1. Introduction

In the current healthcare context, providing quality care is one of the greatest challenges facing healthcare professionals worldwide. Advances in healthcare and technology are leading to overwork, resulting in an increase in errors and a decline in the quality of care [1]. For example, amid multiple demands and insufficient resources, it may be too difficult to meet all nursing care requirements, and nurses may choose to take short-cuts, or delay or simply skip tasks related to patient care, such as posture changes or risk assessment [2].

Clinical documentation is an essential tool for quality assurance, as it facilitates the flow of information between health professionals from different disciplines and work areas [3]. In particular, nursing documentation provides important information about the care of hospitalized patients and is an important indicator of the effective delivery of patient care [4]. Therefore, nursing documentation can be used to evaluate and improve the quality of care in healthcare facilities [5].

Given that nurses' work affects patient outcomes, it is important to create ideal conditions to enable continuity of care and good patient outcomes [6]. To achieve these goals, nursing documentation must contain valid, reliable information that meets certain standards of validity and quality [7]. Specifically, frequency, accuracy, and completeness of the records are considered quality standards for nursing assessments [8]. Only then can nurses

play a key role in promoting cost-effective, high-quality nursing outcomes, improving patient safety and identifying patient needs through nursing assessments [9].

Nowadays, however, nurses perceive documentation to be an administrative burden due to the amount of data and number of elements to be recorded, although clinical history taking is now computerized [10–12]. Nevertheless, several studies [13–15] advocate electronic records over paper format, suggesting that they can improve the quantity of records and the content quality of nursing documentation. In particular, nursing documentation is considered fundamental to individual care planning and nursing assessment in general [16]. However, various studies have shown that nursing assessment recording does not meet the appropriate standards for quality and quantity of information [4,5,17]. All of this has a negative impact on the quality of care and the incidence of adverse effects [18].

Some studies specifically assessed the overall quality of nursing assessments [19,20]. Other studies assessed the quality of nursing records used to assess pain and nutritional status and concluded that these should be improved to ensure quality of care [5,21]. However, no studies were found that assessed the quality of nursing documentation for assessing functional capacity, which is one of the most important determinants of nursing effort [22]. In addition, no studies were found that evaluated documentation in relation to assessing the risk of pressure injuries or falls, two relevant nursing-sensitive outcomes [23].

Due to the exposed gap, the VALENF Project (Nursing Assessment by its acronym in Spanish) was raised [24]. This project arose from the need to improve nursing assessments using a specific nursing model or framework. The instruments to assess functional capacity, the risk of pressure injuries, and the risk of falls are probably the most used by nurses in adult hospitalization units. In this case, they were measured with the Barthel index, the Braden index, and the Downton scale, since they were the instruments used by protocol in the participating hospital. These instruments are used independently, but share constructs, dimensions, and items that are duplicated and redundant. Knowing that these evaluations are the basis for carrying out diagnoses and interventions that adjust to the needs of the patient, the objective of the VALENF Project was to combine the nursing assessment of functional capacity, risk of pressure injuries, and risk of falls in a single an instrument that integrates them. Thus, a more parsimonious seven-item meta-tool has been obtained with a high predictive capacity and reliability compared to the original instruments [25,26].

Consequently, the overall objective of this study was to analyze the impact of a multifaceted strategy within the VALENF Project to improve the assessment of functional capacity, pressure injury risk, and fall risk at the time of admission of adult inpatients.

## 2. Materials and Methods

### 2.1. Design and Setting

A secondary analysis of the VALENF project databases was carried out [24]. This project was carried out at the Hospital Universitario de La Plana (Spain). This is a public hospital serving about 200,000 residents.

### 2.2. Participants and Sample

The study was conducted using two databases from the VALENF project. The first database included nursing assessments conducted between October and December 2020, before the implementation of the strategy to promote improvements in nursing assessment. The second database included nursing assessments conducted between October and December 2021, after the implementation of the strategy. In both databases, the study population consisted of nursing assessments performed on patients admitted to adult inpatient units at the participating hospital.

To assess the number of completed records, nursing assessments of patients over 18 years of age admitted to adult inpatient units were considered. Assessments of patients admitted to the intensive care unit were excluded because they were subject to a different protocol and instruments for assessment at admission to the unit, as were assessments of patients admitted to the home care unit, maternal–infant unit, and obstetrics and gynecol-

ogy unit. Nursing assessments of patients transferred from other departments or hospitals were also excluded.

After the initial screening, the quality of the records was assessed. For this purpose, we considered nursing assessments performed in the first 24 h after admission, which included a complete assessment of functional capacity, risk of pressure injury, and risk of falls. Records that did not meet the inclusion criteria were excluded.

Description of the Sample Selection Process

A total of 5262 nursing assessments were initially reviewed for inclusion in the study, 48.7% ($n = 2561$) belonging to the period of 2020 and 51.3% ($n = 2700$) to 2021. First, of those belonging to the intensive care unit, 1.56% ($n = 40$) and 1.76% ($n = 45$) were from the obstetrics–gynecology unit and 1.05% ($n = 27$) were excluded because they were transfers in the 2020 period. In 2021, 3.37% ($n = 91$) were excluded because they belonged to the intensive care unit, 5.07% ($n = 137$) because they were from the obstetrics–gynecology unit, and 0.41% ($n = 11$) because they were transfers. Thus, a total of 2449 and 2460 assessments were included to analyze the number of assessments for the periods of 2020 and 2021, respectively. Of the 2449 included in the 2020 period, 12.98% ($n = 318$) were excluded because they were completed more than 24 h after admission or did not have a completion date, and 59% ($n = 1445$) were excluded because they were not completed correctly. The final sample consisted of 686 (28.01%) nursing assessments for the 2020 period. In the second data collection period, from October to December 2021, of the 2460 assessments collected, the main reason for exclusion was the noncompletion of any of the scales in 36.21% ($n = 891$), followed by late completion (>24 h after admission) or not submitting the date of completion, which accounted for 8.29% ($n = 204$) of the sample. A final sample of 1445 (58.73%) remained for the 2021 period (Figure 1).

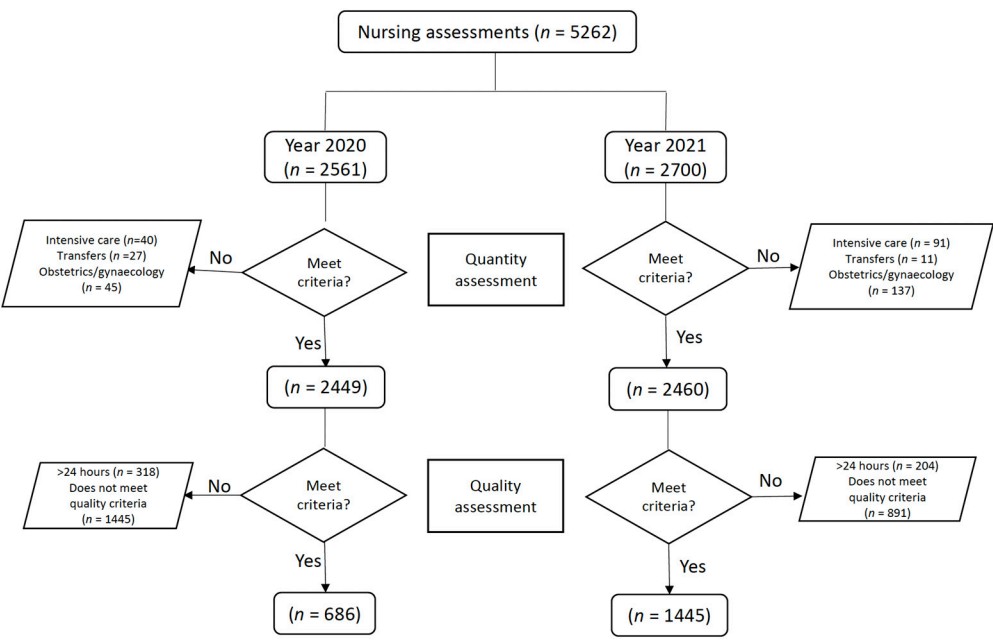

**Figure 1.** Sample selection process.

*2.3. Variables and Instruments*

2.3.1. Variables Related to Nursing Assessment

The following nursing assessment instruments were included in the VALENF Project, since they are the instruments used in the center by protocol:

1. Barthel index: This assesses the functional capacity (or dependency level) to carry out basic activities of daily life. It comprises 10 items, with a total score range between 0 and 100, and groups the patients into four levels (low dependency > 60;

moderate dependency = 40–55; severe dependency = 20–35 points; total dependency = 0–15 points) [27]. The Barthel index was validated for use in Spain by González et al. [28], showing adequate internal consistency (Cronbach's alpha > 0.8) and construct validity (RMSEA < 0.08; LI > 0.9).

2. Braden index: This assesses the risk of pressure injuries. It comprises six items. Its scores range from six to 23 points, and it is classified into four categories (no risk = 19–23; low risk = 15–18; moderate risk = 13–14; high risk = 6–12 points). The Braden index is a widely used and validated instrument internationally and in Spain, always showing good diagnostic accuracy indicators (sensitivity = 0.27–1; specificity = 0.26–0.92; positive predictive value = 0.08–0.77; negative predictive value = 0.71–1) [29].

3. Downton scale: This assesses the risk of falls and comprises five items that score zero or one points. Higher scores indicate higher risk of falls, and scores above two points indicate a high risk of falls. Bueno-García et al. [30] studied the diagnostic accuracy of the Downton scale in Spain, obtaining a sensitivity of 0.58 points and a specificity of 0.62 points.

In addition, the hospitalization unit where the admission occurred was recorded. On the one hand, the quantity of completed assessments was assessed by the absence or presence of completed assessment instruments. Assessments that were not recorded were considered absent. On the other hand, the assessment of quality of completion was assessed whether these nursing assessment instruments were completed in the first 24 h after admission (nursing assessment protocol of the center) and whether information deficits were present in the completed nursing records. Deficiencies were considered to be those records whose information was inadequate or inaccurate when the received records were reexamined (when some features of the assessment instruments were not recorded or were omitted: Barthel, Braden, or Downton). For example, instrument categorization in the hospital's electronic record system is an element that is entered manually. Therefore, it was assumed that any scale that was not completed correctly was not of sufficient quality to be considered.

2.3.2. Strategy to Improve the Assessment of Functional Capacity, the Risk of Pressure Injuries, and the Risk of Falls

The overall goal of the VALENF project [24] was to develop and validate a meta-instrument that integrates assessment of functional capacity, pressure injury risk, and falls risk with a new approach to nursing assessment in adult hospitalization units [25,26]. One of the limitations of the project was the lack of validity and reliability of the data due to incomplete, inconsistent, or inaccurate assessments [31,32].

To overcome this limitation, a multifaceted strategy [33] was developed to improve the assessment of functional capacity, fall risk, and pressure injury risk that nurses perform on patients when they are admitted to the adult hospitalization units. This strategy was adapted to the different levels of care management [34]. As a starting point, it should be said that the center had a nursing assessment protocol that specified that nursing assessment included the three instruments used in this study and that it should be performed within the first 24 h after admission. It was also intended to clarify that the performance of the nursing assessment was considered a quality objective that would be used at the macromanagement level to assess nursing performance in this study [35].

Initially, meetings were scheduled by nursing directors to remind nurses of the nursing assessment protocol when admitting patients to the adult hospitalization units. In addition, the need to reduce variability in the execution of this protocol was emphasized [36]. Until then, the supervisors of some units undertook this task and performed the functional capacity assessment at the time of discharge rather than in the first 24 h after admission.

Second, the supervisors of the hospitalization units participating in the project received detailed information about the correct functioning of the assessment instruments included in the study by reminding them about the functioning of the electronic medical record

software [8]. This was an attempt to break the general perception of low utility in daily clinical practice [37,38]. In addition, the intention was conveyed to increase the completion of nursing assessments and to increase the efficiency of duty nurses.

Third, informal visits were made to nurses in their workplaces to explain the strategy and purpose of the VALENF project and the improvements that would incorporate the results into daily clinical practice. To ensure that information was shared vertically and horizontally [39], a champions team of nurses working in these departments was also formed [40] to emphasize the goals of the project and the importance of appropriate nursing assessment at patient admission.

### 2.4. Data Collection

In both periods, data were obtained from the center's electronic medical records. The nurses of the included hospitalization units performed the data collection as part of their routine work [5]. Pseudonymized databases were requested from the hospital's documentation service, without including any personal data that would allow users to be identified. Previously, a consensus was reached with the documentation and informatics services on the structure of the databases.

### 2.5. Data Analysis Procedures

First, we described the nursing assessments recorded in the adult hospitalization units during both data collection periods (October–December 2020; October–December 2021). Differences in the number of records completed in each period, both overall and by hospitalization unit, were analyzed using the chi-squared test ($X^2$).

Second, we analyzed whether there were significant differences in the quality of completion of the instruments for functional capacity, risk of falls, and risk of pressure injuries in each period and in each of the hospitalization units. To this end, we studied whether or not the quality of the assessment of the three instruments as a whole and each instrument separately (Barthel index, Braden index, and Downton scale) was met with a chi-squared test ($X^2$). A value of $p < 0.05$ was considered in the hypothesis contrasts and the statistical analysis was performed with Jamovi v. 2.3.2 software.

### 2.6. Ethical Considerations

This study was accepted by the manager of the participating hospital and positively evaluated by the Ethics and Research Committee in December 2020 (code VALENF. 9 December 2020). The study has been designed in accordance with Regulation (EU) 2016/79 of the European Parliament and of the Council of 27 April 2016, regarding the protection of natural personas and the Organic Law 3/2018, of 5 December, about Personal Data Protection and Guaranteeing Digital Rights, as specifically indicated by its additional 17th disposition, section (d), which considers the lawful use of pseudonymized personal data for health research purposes, particularly for biomedicine. Therefore, the Ethics and Research Committee approved the request for an exemption from informed consent.

## 3. Results

### 3.1. Quantity of Completed Assessments

Of the 2449 records included in 2020, 28.01% ($n = 686$) were completed. In 2021, of the 2460 assessments included, 58.73% ($n = 1445$) were completed, with a significant difference between the two periods with respect to the percentage of records included ($p < 0.001$) (Table 1). In the 2020 period, the trauma (69.67%; $n = 301$) and internal medicine (89.43%; $n = 237$) units had the highest percentage of completion, with the same units in the 2021 period (trauma = 85.17%; $n = 380$, internal medicine = 93.3%; $n = 195$). However, the percentage of compliance increased significantly in all hospitalization units, except in the general surgery ward, with 21.61% ($n = 78$) in the first period and 10.83% ($n = 38$) in the second ($p < 0.001$).

**Table 1.** Quantity completed for the hospitalization unit.

| | 2020 | | 2021 | | |
| | Total | Quantity | Total | Quantity | |
| **Hospitalization Unit** | $n^1$ | % (*n*)[2] | *n* | % (*n*) | $p^3$ |
|---|---|---|---|---|---|
| Traumatology | 432 | 69.67 (301) | 441 | 85.17 (380) | <0.001 |
| Surgery and gynecology | 396 | 7.32 (29) | 377 | 78.77 (297) | <0.001 |
| Cardio/gastroenterology | 234 | 1.28 (3) | 330 | 66.36 (219) | <0.001 |
| Neuro/pulmonology | 322 | 10.87 (35) | 311 | 61.73 (192) | <0.001 |
| General surgery | 355 | 21.97 (78) | 350 | 10.85 (38) | <0.001 |
| Otolaryngology/urology | 455 | 0.66 (3) | 443 | 27.99 (124) | <0.001 |
| Internal medicine | 265 | 89.43 (237) | 209 | 93.30 (195) | <0.001 |
| TOTAL | 2449 | 28.01 (686) | 2460 | 58.73 (1445) | <0.001 |

[1] *n*: sample; [2] percentage (sample); [3] $X^2$.

### 3.2. Quality of Completion of Assessment Instruments

In the period leading up the implementation of the strategy, 2020, 63.4% (*n* = 435) of the cases met the quality criteria established to consider adequate completion of the three instruments. This percentage then increased to 71.8% (*n* = 1038), a statistically significant difference (*p* < 0.001). Regarding the analysis based on the instruments, in 2020, the Barthel index obtained the highest percentage of completion quality, with 89.2% (*n* = 612) of the cases, followed by Downton (84.1%; *n* = 577) and Braden (83.5%; *n* = 573). In 2021, the Barthel index maintained first place, with 91.2% (*n* = 1318), followed by Braden (89.7%; *n* = 1296) and, finally, Downton (87%; *n* = 1257). In all three instruments, the quality of completion increased in the second data collection period, but only the Braden instrument showed significant differences when comparing both samples at the global level (*p* < 0.001) (Table 2).

In addition, Table 2 shows that the percentage of completion of the records in the three instruments as a whole increased in the hospitalization units in the second data collection period, although there were no significant differences in any unit. The surgery and gynecology hospitalization unit had the greatest increase in complete assessments, going from 75.9% (*n* = 22) to 80.8% (*n* = 240) in 2021, although it was the unit that registered the fewest assessments.

The Barthel index obtained a global percentage of quality of compliance of 89.2% (*n* = 612) in 2020 and 91.2% (*n* = 1318) in 2021, with no significant differences (*p* = 0.140). The percentage of quality of completion increased, although only the neuro/pulmonology unit showed significant differences, from 82.9% (*n* = 29) to 94.3% (*n* = 181). In addition, surgery and gynecology, traumatology, otolaryngology/urology, and internal medicine had no significant difference in any of them.

The global quality of completion of the Braden index improved significantly, from 83.5% (*n* = 573) to 89.7% (*n* = 1296) (*p* < 0.001). The quality of completion in each of the hospitalization units also improved, except in the general surgery unit from 84.65% (*n* = 66) to 97.4% (*n* = 37) and internal medicine from 78.5% (*n* = 186) to 86.2% (*n* = 168), showing significant differences (*p* < 0.005).

Finally, the percentage of quality of Downton scale completion improved in the seven hospitalization units, being 84.1% (*n* = 577) in 2020 and 87% (*n* = 1257) in 2021, with no significant differences (*p* = 0.073). All units improved the percentage of the Downton scale recording quality, where the surgery and gynecology unit obtained the highest percentage with 93.3% (*n* = 277) in 2021, although there was no significant difference in any case (*p* > 0.05).

**Table 2.** Quality completion by hospitalization unit.

| Hospitalization Units | | Overall Quality | | | Barthel Quality | | | Braden Quality | | | Downton Quality | | |
|---|---|---|---|---|---|---|---|---|---|---|---|---|---|
| | | 2020 | 2021 | | 2020 | 2021 | | 2020 | 2021 | | 2020 | 2021 | |
| | | % (*n*) [1] | % (*n*) [1] | *p* [2] | % (*n*) | % (*n*) | *p* | % (*n*) | % (*n*) | *p* | % (*n*) | % (*n*) | *p* |
| Traumatology | Yes | 69.8 (210) | 71.3 (271) | 0.660 | 93 (280) | 93.2 (354) | 0.945 | 86.4 (260) | 87.4 (332) | 0.704 | 86.4 (260) | 87.1 (331) | 0.781 |
| | No | 30.2 (91) | 28.7 (109) | | 7 (21) | 6.8 (26) | | 13.6 (41) | 12.6 (48) | | 13.6 (41) | 12.9 (49) | |
| Surgery and gynecology | Yes | 75.9 (22) | 80.8 (240) | 0.522 | 93.1 (27) | 92.6 (275) | 0.920 | 86.2 (25) | 90.6 (269) | 0.451 | 86.2 (25) | 93.3 (277) | 0.165 |
| | No | 24.1 (7) | 19.2 (57) | | 6.9 (2) | 7.4 (22) | | 13.8(4) | 9.4 (28) | | 13.8 (4) | 6.7 (20) | |
| Cardio/ gastroenterology | Yes | 33.3 (1) | 68.9 (151) | 0.187 | 66.7 (2) | 93.6 (205) | 0.065 | 66.7 (2) | 86.8 (190) | 0.312 | 66.7 (2) | 84 (184) | 0.418 |
| | No | 66.7 (2) | 31.1 (68) | | 33.3 (1) | 6.4 (14) | | 33.3 (1) | 13.2 (29) | | 33.3 (1) | 16 (35) | |
| Neuro/ pulmonology | Yes | 57.1 (20) | 72.9 (140) | 0.060 | 82.9 (29) | 94.3 (181) | 0.018 | 88.6 (31) | 94.3 (181) | 0.212 | 74.3 (26) | 82.8 (159) | 0.232 |
| | No | 42.9 (15) | 27.1 (52) | | 16.6 (6) | 5.7 (11) | | 11.4 (4) | 5.7 (11) | | 25.7 (9) | 17.2 (33) | |
| General surgery | Yes | 67.9 (53) | 81.6 (31) | 0.123 | 96.2 (75) | 84.68 (33) | 0.063 | 84.6 (66) | 97.4 (37) | 0.041 | 84.6 (66) | 92.1 (35) | 0.259 |
| | No | 32.1 (25) | 18.4 (7) | | 3.8 (3) | 15.32 (5) | | 15.4 (12) | 2.6 (1) | | 15.4 (12) | 7.9 (3) | |
| Otolaryngology/ urology | Yes | 66.7 (2) | 77.4 (96) | 0.661 | 100 (3) | 92.7 (115) | 0.628 | 100 (3) | 96 (119) | 0.723 | 66.7 (2) | 85.5 (106) | 0.367 |
| | No | 33.3 (1) | 22.6 (28) | | 0 (0) | 7.3 (9) | | 0 (0) | 4 (5) | | 33.3 (1) | 14.5 (18) | |
| Internal medicine | Yes | 53.6 (127) | 55.9 (109) | 0.631 | 82.7 (196) | 79.5 (155) | 0.394 | 78.5 (186) | 86.2 (168) | 0.039 | 82.7 (196) | 84.6 (165) | 0.593 |
| | No | 46.4 (110) | 44.1 (86) | | 17.3 (41) | 20.5 (40) | | 21.5 (51) | 13.8 (27) | | 17.3 (41) | 15.4 (30) | |
| TOTAL | Yes | 63.4 (435) | 71.8 (1038) | <0.001 | 89.2 (612) | 91.2 (1318) | 0.140 | 83.5 (573) | 89.7 (1296) | <0.001 | 84.1 (577) | 87 (1257) | 0.073 |
| | No | 36.6 (251) | 28.2 (407) | | 10.8 (74) | 8.8 (127) | | 16.5 (113) | 10.3 (149) | | 15.9 (109) | 13 (188) | |

[1] Percentage (sample) of instruments correctly completed; [2] $X^2$.

## 4. Discussion

### 4.1. Importance of Assessing Functional Capacity, Risk of Falls, and Pressure Injuries

The literature indicates that computerized nursing records speed up the assessment process when complete and accurate documentation is available [41]. In this case, the availability of computerized nursing records allowed us to access a large amount of data (5262 records). The results of our study show a significant improvement in the quantity of records completed after the strategy, as well a modest improvement in the quality of completed nursing documentation. Such improvement is encouraging for nurses, as the data obtained from nursing assessments can be used as indicators of nursing care and the nursing process [5]. In addition, optimal quality in completing nursing assessment instruments can describe and achieve desired patient outcomes [8].

The importance of assessing the functional capacity, the risk of pressure injuries, and the risk of falls in hospitalized patients arises because they are three sensitive nursing results [42]. In addition, it is important that the evaluations are well carried out since pressure injuries are mostly nosocomial according to the fifth national study in Spain on their prevalence [43]. Functional capacity is one of the main determinants of care intensity [22,44]. Falls in the hospital environment have an incidence between 0.6% and 14.3% in Spain [45]; in 2017, it was the most reported adverse event in hospitals [46]. The presence of these three indicators is related to a higher risk of mortality [18].

### 4.2. Factors That Influence the Quality of Nursing Assessment

Although technophobia and resistance to technology have been blamed, until recently, nurses have had minimal influence on the design of electronic health records. Therefore,

some contend that nurses questioning systems that do not meet the documentation requirements of nursing practice [6] best explains the perceived resistance to computerized records. The need to monitor quality of care is very important to hospital management [5], so it is recommended that organizations have a policy of reviewing electronic record systems prior to implementation. In our study, the strategy was developed within the structured evaluation frameworks mentioned by Munroe et al. [47], which improves the clinical performance of patient assessment.

The main reason for excluding records was that they did not meet the selection criteria, mainly due to inadequate completion of the assessment instruments, in accordance with previous studies [4,48]. It is important to mention that a high percentage of records were excluded because, in the period before the implementation of the strategy, the Barthel instrument was completed for patient discharge and not at admission, as required by the hospital protocol and the selection criteria in the present study. In addition, in agreement with other authors [49,50], we found that nurses used informal screening approaches instead of the numeric rating scale, which resulted in judgment based on their own clinical judgment and was not always consistent with the results of the assessment instruments.

The variability in the percentages of the assessments included depending on the different hospitalization units is a noteworthy aspect, even though the hospital has a standardized nursing assessment protocol. It is possible that these differences are due to high staff turnover, differences in organizational model of the unit, leadership, or supervisory influence [51]. However, further research is needed to confirm these possible explanations. Kalisch et al. [52] suggest that the main reason for inadequate completion of nursing documentation is work-related aspects such as the need for human resources or lack of time. An example of this may be the situation during the first wave of the COVID-19 pandemic. During this time, nurses spent an average of 10 to 15 min donning personal protective equipment to care for patients, which increased their workload. In order to reduce this work overload, nurses were instructed not to carry out nursing assessments on admission, and so it is suspected that reversing these instructions would be a slow process. Because of this and the fact that the data were taken from the electronic medical record, there is a potential bias in the data regarding the quantity and quality of nursing assessments, as has been noted in other studies [31,32,53].

Nurses not completing nursing documentation according to appropriate quantitative and qualitative standards can compromise the validity of nursing assessment and the identification of high-risk patients [54]. Assessment of nursing record completeness has been the subject of numerous studies [5,8,49,55]. However, no published articles have been found on the quality of nursing assessments that include assessment of functional capacity, risk of pressure injury, and risk of falls at the time of admission to an adult inpatient facility, so it is difficult to agree on an appropriate standard for the quality of nursing records that assesses these dimensions.

Several authors [5,49,55] have suggested that a variety of factors affect the quality of record completion, such as the amount of care provided by the nurses who perform the records, the characteristics of the nurses, the level of education and training of the nurses, years of work, age, or greater contact with patients. There is a lack of training to support the use of instruments in the nursing profession. Palese et al. [54] found that only 58.6% of nursing supervisors and 57.1% of clinical nurses reported specific training opportunities on instruments in their professional training, which may affect their accuracy in daily use. It is likely that more educational support for nurses will be needed in the future, although Anthony et al. [56] noted that the interplay of education, clinical judgment, and assessment tool use has not been fully explored. Future studies could focus on examining. All this leads to important indications of concern about the analysis of the quality of records. This is reflected in several studies, such as those by Muinga et al., Silva de Melo et al., Gaedke Nomura et al., and Akhu-Zaheya et al. [3,20,57,58]. These studies highlight the need for pedagogical interventions that focus on improving the quality of records. Therefore, this

study provides a starting point for developing strategies to improve record keeping in our context.

The improvement in completion after employing the multifaceted strategy is noticeable in terms of the number of completed assessments, and discrete in terms of the quality of completion of the assessment instruments. It should be noted, however, that the effect of this type of strategy diminishes over time, so that it is suspected that a behavioral change of this nature may take years to achieve its effectiveness [33]. Other authors advocate that the use of an audit instrument in combination with other types of interventions, such as the strategy uses, could be useful for measuring the quality of nursing documentation [59].

Therefore, further studies may be useful to investigate the impact of nursing care on the quantity and quality of nursing documentation, as well as the creation of standards to classify this quality. Likewise, it would be useful to study the influence that the adaptation of the communication style used at each level of care had on the results of the quality and quantity of the records [60]. The data collected in our study did not allow us to evaluate in depth the motivations that led to this result.

### *4.3. Limitations*

The present study has some potential limitations. For example, the data were collected at a single hospital, which may raise concerns about the generalizability and comparability of the results. In addition, a sample size calculation was not performed as the data were obtained from the database used for VALENF Instrument Development [25].

It should be noted that the strategy used can be improved and considered as an educational intervention that would allow measuring other results, such as the effectiveness or the level of knowledge of the nurses.

However, this study has important implications for clinical practice, especially for improving nursing care in patients admitted to inpatient units. Improved nursing records may facilitate the detection of at-risk individuals and provide more individualized care. In addition, improved records may increase nursing time for direct patient care [12]. The results of this study present an opportunity for nurse leaders to implement structured interventions aimed at improving compliance with nursing records. This may be the beginning of a paradigm and attitude shift, and an important step in improving the culture of quality in nurses' clinical practice.

## 5. Conclusions

The results support the use of strategies such as the one used in this study to improve the assessment of functional capacity, the risk of pressure injuries, and the risk of falls. Nursing assessments improved in both quantity and quality, although there were differences depending on the adult hospitalization units and the instruments. These results represent progress in improving the quality of clinical records and may help improve the quality of patient care in the early detection of risks.

**Author Contributions:** Conceptualization, D.L.-A. and I.L.-R.; methodology, V.M.G.-C.; formal analysis, I.L.-R., D.L.-A. and V.M.G.-C.; investigation, M.J.V.-C., R.M.-G. and G.M.-M.; data curation, I.L.-R.; writing—original draft preparation, I.L.-R. and D.L.-A.; writing—review and editing, V.M.G.-C. and G.M.-M.; project administration, D.L.-A. and V.M.G.-C.; funding acquisition, V.M.G.-C. All authors have read and agreed to the published version of the manuscript.

**Funding:** This research was funded by Universitat Jaume I, grant number UJI-A2020-08.

**Institutional Review Board Statement:** The study was conducted in accordance with the Declaration of Helsinki and approved by the Institutional Review Board of Hospital Universitario de La Plana (code VALENF. 9 December 2020).

**Informed Consent Statement:** Patient consent was waived due to this being a retrospective study based on recorded data.

**Data Availability Statement:** All necessary data are supplied and available in the manuscript; however, the corresponding author will provide the dataset upon request.

**Public Involvement Statement:** There was no public patient participation.

**Guidelines and Standards Statement:** This manuscript was drafted against the STROBE guidelines for observational research [61].

**Conflicts of Interest:** The authors declare no conflict of interest.

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
