# Peer review of "Improving Nursing Assessment in Adult Hospitalization Units: A Secondary Analysis"

_nursrep, doi:10.3390/nursrep13030099_

Round 1

Reviewer 1 Report

Dear authors, congratulations for the work done. The submitted manuscript shows great interest in assessing the quality of nurses' records using the Barthel, Braden and Downton scales on admission to hospital; this allows to evaluate improvements in the nurses' records before and after implementing a strategy to assess the functional capacity, risk of pressure injuries, and  risk of falls.

Introduction: Places the reader in the field of research in the correct way.

Methods: I should only point out a possible weakness of the study due to the non-estimation of the sample size calculation, although it is understood that this population size corresponds to that reached in previous studies; however, some statistical data on the proportion of the population reached in previous studies could be indicated in the sub-heading participants and sample.

Results: Well-structured and with sufficiently detailed information.

Discussion and conclusions: Appropriate.

References: Right.

Author Response

Dear reviewer:
Thank you for your feedback. We attach a document where we indicate the corrections and changes made.

Reviewer 2 Report

Abstract: Adequate.

Introduction: Sufficient for understanding the study's problem, however in view of the stated objective it is not clear if what they want is to evaluate the documentation or the impact of the strategy that they refer to, but do not describe. This clarification would benefit from being improved, as the data come from a larger study.

Method: In line with the introduction, the VALENF project must be explained. It is not understood what its scope is, the data used appear without framework. 2.3.1 Variables and instruments: It is not clear why these Scales. 2.3.2. Strategy to improve the assessment of functional capacity, the risk of pressure inju-11 ries and the risk of falls: The stated objective of the VALENF project still does not allow us to understand to what extent the Scales used contributed to the construction of a global instrument.

Results: Point 3.1. Description of the sample selection process should integrate the method, refers to data collection. As the implemented strategy is not known, the results are not understood, why did some improve and others did not, what does it mean?

Discussion: It focuses on a line that is not coherent with the results and the reference to the instruments used, the documentation in nursing. It is not important to know or reflect in this study only on nursing documentation, but also on the specificity of what was valued and recorded with these instruments and their added value. Confused and inconsistent.

Conclusions: Brief and in line with what has already been mentioned.

Author Response

(The authors gave the same response as above.)

Reviewer 3 Report

Thanks for letting me review this manuscript. What follows is a series of recommendations to help improve the manuscript:

1) In the abstract, please add the type of strategy used and the type of quality of assessment that was meant to improve

1) Intro. It is not clear to me wha the authors mean by improving the quality of the documentation. It is an important aspect to made be clearer in this section.

2) Materials  and _Methods.  The instrument sections should be written in a more structured manner, for each instrument used. Number of items, score range, likert scale, etc.

3)Results: the authors wrote: In general, 63.4% (n = 435) of the cases prior to the strategy met the quality criteria established to consider adequate completion of the three instruments together. It seems references are lacking for this sentence

4)Discussion. The authors wrote: Several authors [6,44,50] have suggested a variety of factors,  such as the amount of 301 care provided by the nurses who perform the records, the characteristics of the nurses, the 302 level of education and training of the nurses, years of work, age, or greater contact with 303 patients, for example. A variety of factors regarding what?

5) I think the "strategy" should be defined as a real educational intervention. Do the authors have measured other short term effectiveness outcomes of this approach? Such as level of knowledge? Demonstrating that improvement was not limited only to distal outcomes (quality and quantity of documentation), would add more quality to the study.

minor editing required

Author Response

(The authors gave the same response as above.)
